# Lipidation-independent vacuolar functions of Atg8 rely on its noncanonical interaction with a vacuole membrane protein

Xiao-Man Liu[1†], Akinori Yamasaki[2†], Xiao-Min Du[1,3†], Valerie C Coffman[4], Yoshinori Ohsumi[5], Hitoshi Nakatogawa[6], Jian-Qiu Wu[4], Nobuo N Noda[2*], Li-Lin Du[1*]

[1]National Institute of Biological Sciences, Beijing, China; [2]Institute of Microbial Chemistry, Tokyo, Japan; [3]College of Life Sciences, Beijing Normal University, Beijing, China; [4]The Ohio State University, Columbus, United States; [5]Unit for Cell Biology, Institute of Innovative Research, Tokyo Institute of Technology, Yokohama, Japan; [6]School of Life Science and Technology, Tokyo Institute of Technology, Yokohama, Japan

**Abstract** The ubiquitin-like protein Atg8, in its lipidated form, plays central roles in autophagy. Yet, remarkably, Atg8 also carries out lipidation-independent functions in non-autophagic processes. How Atg8 performs its moonlighting roles is unclear. Here we report that in the fission yeast *Schizosaccharomyces pombe* and the budding yeast *Saccharomyces cerevisiae*, the lipidation-independent roles of Atg8 in maintaining normal morphology and functions of the vacuole require its interaction with a vacuole membrane protein Hfl1 (homolog of human TMEM184 proteins). Crystal structures revealed that the Atg8-Hfl1 interaction is not mediated by the typical Atg8-family-interacting motif (AIM) that forms an intermolecular β-sheet with Atg8. Instead, the Atg8-binding regions in Hfl1 proteins adopt a helical conformation, thus representing a new type of AIMs (termed helical AIMs here). These results deepen our understanding of both the functional versatility of Atg8 and the mechanistic diversity of Atg8 binding.
DOI: https://doi.org/10.7554/eLife.41237.001

*For correspondence:
nn@bikaken.or.jp (NNN);
dulilin@nibs.ac.cn (L-LD)

†These authors contributed equally to this work

## Introduction

Macroautophagy (hereafter autophagy) is an evolutionarily conserved bulk degradation pathway essential for cellular homeostasis. In autophagy, cytosolic materials to be degraded are sequestered and enclosed inside double-membrane vesicles termed autophagosomes. Autophagosome formation is a complicated process that requires many autophagy-related (Atg) proteins (*Mizushima et al., 2011*). Among them, the ubiquitin-like protein Atg8 (called LC3/GABARAP proteins in humans) plays a central role. Atg8 is conjugated to the membrane lipid phosphatidylethanolamine (PE) through a series of enzymatic reactions including its processing by the protease Atg4, activation by an E1-like enzyme Atg7, and covalent linking of its G116 residue to PE by the collaborative activities of an E2-like enzyme Atg3 and an E3-like complex Atg12–Atg5-Atg16 (*Ichimura et al., 2000*; *Hanada et al., 2007*). This covalent attachment of Atg8 to lipid is termed lipidation. Lipidated Atg8 on autophagic membranes is critically important for autophagosome formation (*Kirisako et al., 1999*), and also serves as a recruitment platform for selective autophagy receptors (*Shintani et al., 2002*).

Selective autophagy receptors and other Atg8-interacting proteins bind to Atg8 using a short linear motif termed Atg8-family-interacting motif (AIM) or LC3-interacting region (LIR) (*Pankiv et al., 2007*; *Ichimura et al., 2008*; *Noda et al., 2008*; *Noda et al., 2010*; *Birgisdottir et al., 2013*). The core consensus sequence of the AIM motif is the four-amino-acid sequence W/F/YxxL/I/V, in which 'x' denotes any amino acid. When bound to Atg8, this motif adopts an extended β strand conformation and forms an intermolecular parallel β-sheet with the β2 strand of Atg8. The conserved aromatic residue (W/F/Y) and hydrophobic residue (L/I/V) in the AIM motif insert into two hydrophobic pockets on Atg8, termed the 'W-site' and the 'L-site', respectively (*Noda et al., 2010*). In addition to these two most conserved residues, acidic residues within or flanking the core consensus sequence also contribute to Atg8 binding by electrostatic interactions with basic residues on Atg8.

Apart from its lipidation-dependent roles in autophagy, Atg8 has been shown to play lipidation-independent roles in both autophagic and non-autophagic processes. The lipidation-independent roles that have been discovered in model yeast species are those related to the vacuole, an equivalent of the lysosome in animals. Yeast vacuoles are usually spherical or near-spherical in shape. In the methylotrophic yeast *Pichia pastoris*, the loss of Atg8 caused aberrant vacuole morphology, a phenotype not shared by *atg7Δ* and *atg8-G116A*, two mutants defective in Atg8 lipidation (*Mukaiyama et al., 2004*; *Tamura et al., 2010*). In another model yeast species, the fission yeast *Schizosaccharomyces pombe*, the loss of Atg8 was also shown to cause abnormal vacuole morphologies, including the formation of tubular-shaped vacuoles, but the lipidation- and autophagy-deficient *atg8-G116A* mutant did not exhibit any vacuole morphology abnormalities (*Mikawa et al., 2010*). Lipidation-independent functions of Atg8 have also been reported in animals (*Al-Younes et al., 2011*; *Calì et al., 2008*; *Chang et al., 2013*; *Reggiori et al., 2010*; *Sharma et al., 2014*). The molecular mechanisms underlying the lipidation-independent functions of Atg8 remain unclear.

In this study, we found that in both *S. pombe* and *Saccharomyces cerevisiae*, Atg8 physically interacts with Hfl1, a vacuole membrane protein. This interaction promotes the localization of Atg8 at the vacuole membrane in a lipidation-independent manner. *hfl1Δ* and *atg8Δ* caused the same vacuole-related phenotypes, which are not shared by other *atg* mutants. We solved the crystal structures of Atg8 in complex with Atg8-interacting regions of Hfl1 proteins. The structures showed that the Atg8-Hfl1 interaction is mediated by noncanonical mechanisms—using Hfl1 sequences adopting helical conformation (termed helical AIMs here) and in the case of budding yeast, involving a previously unreported binding site on Atg8. These results unveil the molecular basis of the lipidation-independent vacuolar functions of Atg8 and expand our understanding of the diversity of Atg8-binding mechanisms.

## Results

### Atg8 interacted with Hfl1 and was recruited to the vacuole membrane by Hfl1

To identify Atg8-binding proteins in *S. pombe*, we performed affinity purification coupled with mass spectrometry analysis of C-terminally YFP-FLAG-His$_6$ (YFH)-tagged Atg8 expressed in *atg4Δ* cells (without the Atg4 protease, Atg8-YFH cannot be processed and lipidated). Among the proteins that specifically co-purified with Atg8-YFH is a protein called Hfl1 (systematic ID SPAC30D11.06c) (*Supplementary file 1*). Hfl1 was found in an unpublished study to be a vacuole membrane protein whose absence resulted in abnormal vacuole morphology (the name stands for 'Has Fused Lysosomes') (*Lilavivat, 2013*). It belongs to a conserved eukaryotic protein family (InterPro ID: IPR005178; PFAM ID: PF03619, formerly DUF300). Like other proteins in this family, Hfl1 is predicted to contain seven transmembrane helices followed by a C-terminal cytosolic tail (*Figure 1A* and *Figure 1—figure supplement 1A*). In the model yeasts *S. pombe*, *S. cerevisiae*, and *P. pastoris*, there is one protein belonging to this family per species, whereas in other fungi and in animals, there are often two or more proteins belonging to this family per species (*Figure 1—figure supplement 1B*). In humans, members of this family include three TMEM184 proteins (TMEM184A, TMEM184B, and TMEM184C) and SLC51A (also known as organic solute transporter subunit alpha or OSTα). The TMEM184 proteins are more closely related to the fungal members of this family than SLC51A is (*Figure 1—figure supplement 1B*).

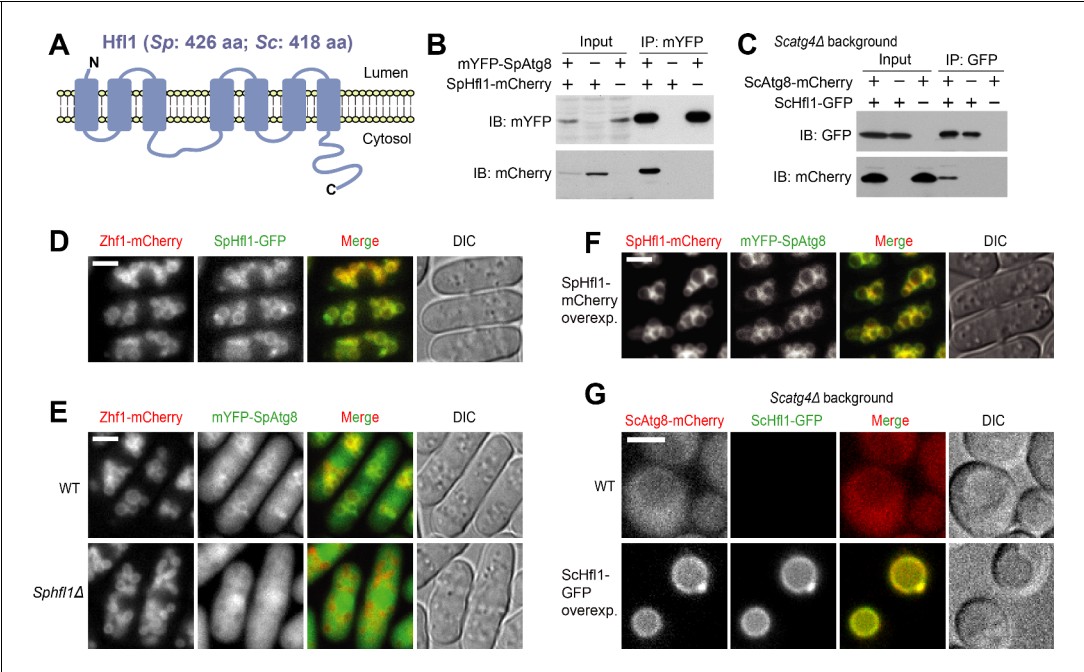

**Figure 1.** Hfl1 interacts with Atg8 and recruits Atg8 to the vacuole membrane. (**A**) Schematic depicting the transmembrane topology of Hfl1 and its related proteins, as predicted using PolyPhobius (see *Figure 1—figure supplement 1A* for a sequence alignment). (**B**) SpHfl1 was co-immunoprecipitated with SpAtg8. (**C**) ScAtg8 was co-immunoprecipitated with ScHfl1. *Scatg4Δ* background was used to prevent the processing of ScAtg8-mCherry. (**D**) SpHfl1 localized to the vacuole membrane. Zhf1 is a vacuole membrane marker. (**E**) mYFP-SpAtg8 exhibited both cytosolic and vacuole membrane localizations in the wild-type cells, and the vacuole membrane localization was abolished in *Sphfl1Δ* cells. (**F**) Overexpression of SpHfl1 using the *nmt1* promoter resulted in the concentration of mYFP-SpAtg8 on the vacuole membrane. (**G**) Overexpression of ScHfl1 using the *TEF1* promoter resulted in the concentration of ScAtg8-mCherry on the vacuole membrane. *Scatg4Δ* background was used to prevent the processing of ScAtg8-mCherry. Bars, 3 μm.

DOI: https://doi.org/10.7554/eLife.41237.002

The following figure supplement is available for figure 1:

**Figure supplement 1.** Sequence alignment and phylogenetic tree of Hfl1-related proteins.

DOI: https://doi.org/10.7554/eLife.41237.003

We confirmed that Atg8 interacts with Hfl1 in *S. pombe* using a co-immunoprecipitation analysis (*Figure 1B*). To determine whether this interaction is conserved, we performed a co-immunoprecipitation analysis in *S. cerevisiae*, and found that the homolog of Hfl1 in *S. cerevisiae*, a previously uncharacterized protein YKR051W, can interact with *S. cerevisiae* Atg8 (*Figure 1C*). We named YKR051W Hfl1. Hereafter, we will use SpHfl1 and ScHfl1 to refer to the Hfl1 proteins in these two yeasts, and use SpAtg8 and ScAtg8 to refer to the Atg8 proteins in these two yeasts.

In fission yeast, endogenously C-terminally GFP-tagged SpHfl1 localized to the vacuole membrane (*Figure 1D*). Endogenously N-terminally mYFP-tagged SpAtg8 exhibited both a cytosolic distribution and a faint but detectable vacuole membrane localization (*Figure 1E*). The vacuole membrane localization of mYFP-SpAtg8 was completely abolished in *Sphfl1Δ* cells (*Figure 1E*). Furthermore, when SpHfl1 was overexpressed from a strong *nmt1* promoter, the cytosolic signal of mYFP-SpAtg8 disappeared and the vacuole membrane localization of mYFP-SpAtg8 became much more conspicuous (*Figure 1F*). In budding yeast, we could not detect endogenously tagged ScHfl1 using live cell imaging, presumably because of low abundance. Using a strong *TEF1* promoter to express ScHfl1, we found that like SpHfl1, it also localized to the vacuole membrane (*Figure 1G*). Similar to the situation in fission yeast, the overexpression of ScHfl1 resulted in the relocalization of ScAtg8 from the cytosol to the vacuole membrane (*Figure 1G*). Thus, as a vacuole-membrane-localized integral membrane protein, Hfl1 binds to Atg8 and is able to recruit Atg8 to the vacuole membrane in both yeasts.

## Loss of Hfl1 resulted in the same vacuole defects as those caused by the loss of Atg8

Given the essential role of Atg8 in autophagy, we examined whether Hfl1 is also important for autophagy. Using CFP-SpAtg8 in fission yeast and GFP-ScAtg8 in budding yeast as reporters to monitor autophagy, we found that neither the loss of SpHfl1 in fission yeast nor the loss of ScHfl1 in budding yeast affected starvation-induced autophagy (*Figure 2—figure supplement 1A–C*).

We then investigated the possibility that Hfl1 is involved in the lipidation-independent non-auto-phagic functions of Atg8. In fission yeast, loss of SpAtg8 but not loss of SpAtg8 lipidation caused aberrant vacuole morphologies (*Mikawa et al., 2010*). The most striking *Spatg8Δ* phenotype reported by Mikawa et al. was tubular-shaped vacuoles, which became more frequent upon treatment with the oxidative-stress-inducing agent paraquat. In our hands, tubular-shaped vacuoles were not readily observed in untreated or paraquat-treated *Spatg8Δ* cells (*Figure 2A* and data not shown), possibly owing to differences in experimental details. After testing a number of other stress-inducing conditions, we found that treating cells with the reducing agent dithiothreitol (DTT) resulted in the formation of tubular-shaped vacuoles in *Spatg8Δ* but not wild-type cells (*Figure 2A and B*). DTT may perturb vacuole function either directly by affecting ion channel activities (*Carpaneto et al., 1999*; *Palmer et al., 2001*), or indirectly by triggering the unfolded protein response (UPR) (*Kimmig et al., 2012*; *Guydosh et al., 2017*). This phenotype of *Spatg8Δ* was shared by *Sphfl1Δ* but not by *Spatg1Δ*, *Spatg2Δ*, *Spatg3Δ*, *Spatg4Δ*, *Spatg5Δ*, *Spatg6Δ*, *Spatg7Δ*, and *Spatg8-G116A* (*Figure 2A and B*, and *Figure 2—figure supplement 1D*), indicating that SpHfl1 and SpAtg8 act in a non-autophagic process to maintain normal vacuole morphology.

The vacuole is important for metal homeostasis and mutants defective in vacuole functions often exhibit altered metal sensitivity (*Ortiz et al., 1992*; *Ramsay and Gadd, 1997*). To further explore the vacuole-related function(s) of SpAtg8 and SpHfl1, we examined whether the loss of SpAtg8 or SpHfl1 affects metal sensitivity. Among the metal salts we tested, which include NaCl, KCl, LiCl, $MgSO_4$, $CaCl_2$, $ZnCl_2$, $CoCl_2$, and $MnCl_2$, *Spatg8Δ* and *Sphfl1Δ* cells exhibited stronger sensitivity to $ZnCl_2$, $CoCl_2$, and $MnCl_2$ than the wild type (*Figure 2C* and data not shown). The metal sensitivity phenotype was not shared by the lipidation-defective mutants *Spatg8-G116A*, *Spatg3Δ*, and *Spatg7Δ* (*Figure 2D* and *Figure 2—figure supplement 2*). Notably, the severity of the metal sensitivity phenotype was the same for *Spatg8Δ* and *Sphfl1Δ*, and the double mutant *Spatg8Δ Sphfl1Δ* did not show stronger phenotype than the two single mutants (*Figure 2C and D*, and *Figure 2—figure supplement 2*), demonstrating that SpAtg8 and SpHfl1 act in the same pathway to confer normal metal tolerance.

Compared to *S. pombe* and *P. pastoris*, the lipidation-independent function(s) of Atg8 in *S. cerevisiae* has been less well characterized. It was reported that *Scatg8Δ* but not *Scatg7Δ* was partially defective in hypo-osmotic stress-induced vacuole fusion (*Tamura et al., 2010*). However, we could not detect this *Scatg8Δ* phenotype, perhaps because it is too mild. Another study reported that vacuolar microdomain formation in stationary phase required ScAtg8 but not ScAtg8 lipidation (*Wang et al., 2014*). We were able to reproduce this observation and found that vacuolar microdomains manifested as the reticular Vph1-mCherry pattern were readily observed in wild-type but not *Scatg8Δ* cells at day 1 (D1) of the stationary phase (*Figure 2E and F*). This phenotype of *Scatg8Δ* was shared by *Schfl1Δ* but not by *Scatg1Δ* and *Scatg7Δ* (*Figure 2E and F*), indicating that ScHfl1 and ScAtg8, but not ScAtg8 lipidation or autophagy, are important for vacuolar microdomain formation in cells cultured to D1 of the stationary phase. Given that Hfl1 physically interacts with Atg8 in both yeasts, the mutant phenotype data strongly suggest that Hfl1 is required for the lipidation-independent vacuolar functions of Atg8 in these two species.

## Mapping the SpAtg8-interacting region of SpHfl1 to amino acids 386–409

To dissect the structure-function relationship of SpHfl1, we first examined which regions of SpHfl1, when removed, disrupt the ability of SpHfl1 to recruit SpAtg8 to the vacuole membrane. The N-terminal 7-transmembrane-helix region of SpHfl1, SpHfl1(1-269), when expressed from a plasmid in *Sphfl1Δ* cells, was able to localize properly to the vacuole membrane but failed to recruit SpAtg8 (*Figure 3—figure supplement 1A*), indicating that SpAtg8 recruitment requires the C-terminal cyto-solic tail of SpHfl1. We then examined a series of C-terminally truncated versions of SpHfl1, and

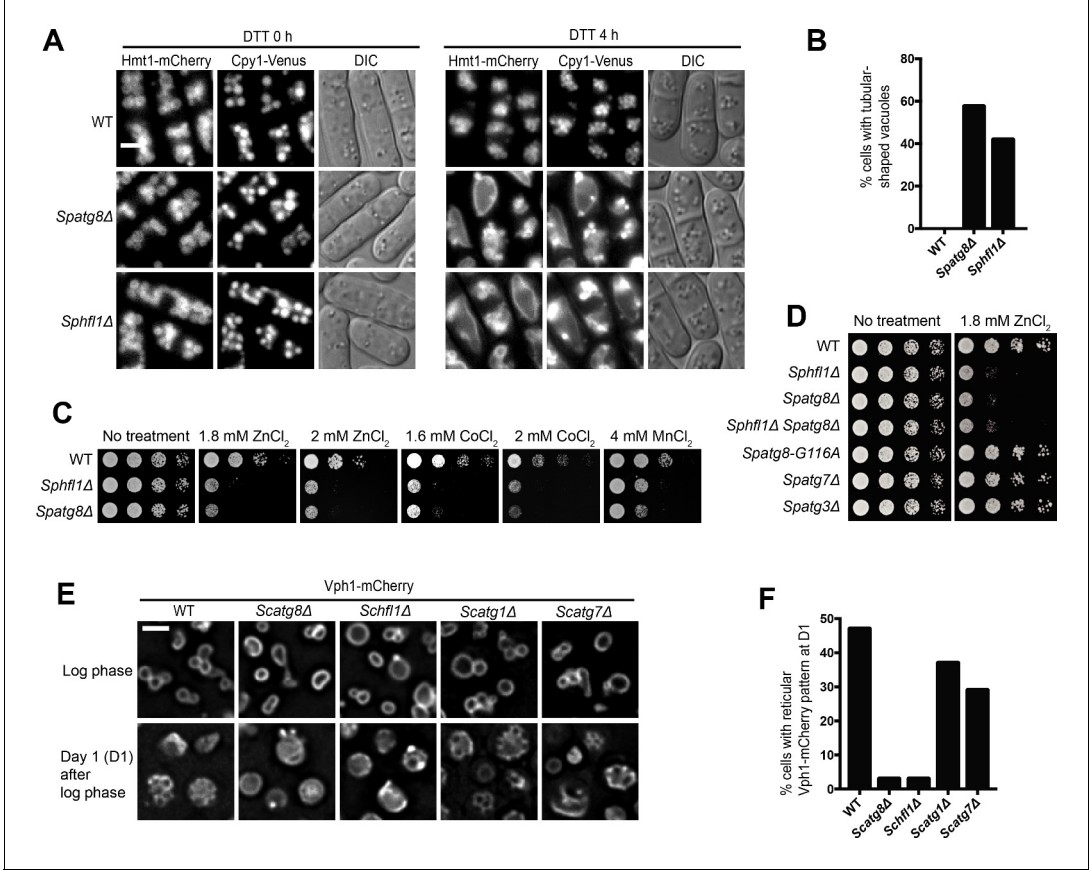

**Figure 2.** Hfl1 is required for the lipidation-independent vacuolar functions of Atg8. (A, B) Micrographs (A) and quantitation (B) showing that DTT treatment induced the formation of tubular-shaped vacuoles in *Spatg8Δ* and *Sphfl1Δ* cells, but not in wild-type cells. Hmt1-mCherry and Cpy1-Venus are a vacuole membrane marker and a vacuole lumen marker, respectively. A representative result of three independent experiments is shown. (C) *Spatg8Δ* and *Sphfl1Δ* exhibited the same metal hyper-sensitivity. Five-fold serial dilutions of cells were spotted on a YES plate and YES plates with metal salts at the indicated concentrations. (D) *Spatg8Δ Sphfl1Δ* double mutant exhibited no enhanced metal sensitivity compared with the two single mutants, and mutants defective in Atg8 lipidation did not show metal hyper-sensitivity. See *Figure 2—figure supplement 2* for the results of $CoCl_2$ and $MnCl_2$ treatment. (E, F) Micrographs (E) and quantitation (F) showing that *Scatg8Δ* and *Schfl1Δ*, but not *Scatg1Δ* and *Scatg7Δ*, were defective in stationary-phase-induced vacuolar microdomain formation. A representative result of three independent experiments is shown. Bars, 3 μm.

DOI: https://doi.org/10.7554/eLife.41237.004

The following figure supplements are available for figure 2:

**Figure supplement 1.** Hfl1 is not required for starvation-induced autophagy and fission yeast autophagy mutants other than *Spatg8Δ* do not exhibit vacuole morphology defect after DTT treatment.

DOI: https://doi.org/10.7554/eLife.41237.005

**Figure supplement 2.** *Spatg8Δ Sphfl1Δ* double mutant exhibited no enhanced metal sensitivity compared with the two single mutants, and mutants defective in Atg8 lipidation did not show metal hyper-sensitivity.

DOI: https://doi.org/10.7554/eLife.41237.006

found that SpHfl1(1-410), which lacks the last 16 amino acids, retained the ability to recruit SpAtg8, whereas SpHfl1(1-385) and several shorter fragments lost the ability (*Figure 3—figure supplement 1A*), indicating that the region between amino acids 385 and 410 is critical for SpAtg8 recruitment. Consistent with this idea, SpHfl1Δ(386-409), which lacks 24 amino acids, also failed to recruit SpAtg8 (*Figure 3—figure supplement 1A*). Using the same set of plasmids to complement the vacuole morphology defect of *Sphfl1Δ*, we found that only SpHfl1(1-410) was able to complement (*Figure 3— figure supplement 1B*), suggesting that the SpAtg8-recruitment ability of SpHfl1 is important for maintaining normal vacuole morphology.

We hypothesized that the recruitment of SpAtg8 to the vacuole membrane by SpHfl1 is mediated by the SpAtg8-SpHfl1 interaction. Supporting this idea, we found that SpHfl1(1-410) but not SpHfl1

(1-385) or SpHfl1Δ(386-409) was able to co-immunoprecipitate SpAtg8 (*Figure 3—figure supplement 2A and B*). Furthermore, the cytosolic tail region, SpHfl1(270-426), was sufficient for co-immunoprecipitating SpAtg8 (*Figure 3—figure supplement 2A*). As Atg8 binding is usually mediated by short sequence segments on Atg8-binding proteins, we proceeded to test whether the 24 amino acids required for SpAtg8 co-immunoprecipitation, SpHfl1(386-409), are sufficient for SpAtg8 binding. We synthesized an SpHfl1(386-409) peptide and performed in vitro pull-down analysis. SpHfl1(386-409) could efficiently pull down recombinant SpAtg8 but not a control protein Yng2-PHD (*Figure 3—figure supplement 2C*). Using the same assay, we found that SpHfl1(386-409) could pull down ScAtg8 as efficiently as SpAtg8 (*Figure 3—figure supplement 2C*), despite that this region of SpHfl1 is not well conserved in ScHfl1 (*Figure 3—figure supplement 2D*).

## Structural basis of the noncanonical interactions between Atg8 and Hfl1

There is no recognizable AIM motif in SpHfl1(386-409) (*Figure 3—figure supplement 2D*), suggesting a novel mode of Atg8 binding. To understand the atomic details of the SpHfl1-SpAtg8 interaction, we solved the crystal structure of SpAtg8 complexed with SpHfl1(386-409) at 2.2 Å resolution (*Figure 3A and B*, *Figure 3—figure supplement 2E*, and *Supplementary file 2*—Table S1). The structure of SpAtg8 is similar to other Atg8-family proteins, consisting of a ubiquitin fold and two α-helices attached at the N-terminus. The structure of SpAtg8 can be superimposed on that of ScAtg8 with an rms difference of 0.8 Å for main-chain atoms except for those of terminal tails (*Figure 3—figure supplement 2F*). As a result, the two AIM-binding hydrophobic pockets (W-site and L-site) on SpAtg8 resemble those on ScAtg8 and other Atg8 homologs (*Figure 3B and D*). SpHfl1(386-409) is comprised of an extended coil followed by an α-helix, and interacts with the W-site, the L-site, as well as α3 of SpAtg8, burying ~2000 Å$^2$ of surface area (detailed interactions are summarized in *Figure 3—figure supplement 3A*). SpHfl1(386-409) forms little interaction with crystallographically adjacent SpAtg8 molecules (*Figure 3—figure supplement 3B*), suggesting that the complex structure is not markedly affected by crystal packing. In the case of canonical AIMs, the consensus W/F/YxxL/I/V sequence (x is any residue) adopts an extended β-conformation and forms an intermolecular β-sheet with β2 of Atg8 (*Figure 3—figure supplement 2G*, **top**), with the side chain of W/Y/F binding to the W-site and that of L/I/V to the L-site. The number of residues intervening the two hydrophobic residues is strictly two (*Figure 3E*). Strikingly, SpHfl1(386-409) neither forms an intermolecular β-sheet with β2 of Atg8 (*Figure 3—figure supplement 2G*, **middle**), nor uses two hydrophobic residues separated by two residues to engage the W-site and the L-site. Instead, SpHfl1(386-409) uses Phe388 at the coil and Tyr398 at the helix for binding to the W-site and the L-site, respectively, with as many as nine residues between them (*Figure 3E and F*). In addition, a hydrophobic interaction is formed between Leu386 of SpHfl1 and a hydrophobic pocket near the W-site, some electrostatic interactions are formed between the side-chains of Asp391 and Glu395 of SpHfl1 and Lys46 and Arg28 of SpAtg8, respectively, and two hydrogen bonds are formed between the main-chain of SpHfl1 and the side-chain of SpAtg8 (*Figure 3B and F*, *Figure 3—figure supplement 3A*).

We next studied the molecular interaction between ScAtg8 and ScHfl1. Using the sequence alignment between SpHfl1 and ScHfl1 as a guide to identify the ScAtg8-binding region in ScHfl1, our GST pulldown analysis indicated that residues 368–389 of ScHfl1 correspond to a minimal region for a strong binding with ScAtg8 (*Figure 3—figure supplement 2H*). Because a complex formed by ScAtg8 and ScHfl1(368-389) did not crystallize, we resorted to a fusion between them for crystallization, a strategy widely utilized for crystallization of a complex between an Atg8-family protein and a canonical AIM (*Suzuki et al., 2014*; *Wu et al., 2015*). The crystal structure of ScHfl1(368-389)-ScAtg8 fusion protein was determined at 2.45 Å resolution (*Figure 3—figure supplement 2I* and *Supplementary file 2*—Table S1). The asymmetric unit of the crystal contains seven copies of the fusion protein, all of which form an intermolecular ScAtg8-ScHfl1(368-389) complex in a head-to-tail manner (*Figure 3—figure supplement 2J*). Among the seven copies of ScHfl1(368-389), two have a slightly different conformation from the others due to crystal packing, while the other five copies have almost the same conformation (*Figure 3—figure supplement 2K and L*); therefore, we use one representative structure of the ScAtg8-ScHfl1(368-389) complex from the five copies hereafter (*Figure 3C and D*). ScHfl1(368-389) has an elongated conformation with a short helical conformation at the residues 374–377, and forms extensive interactions with the W-site, the L-site, α3, as well as

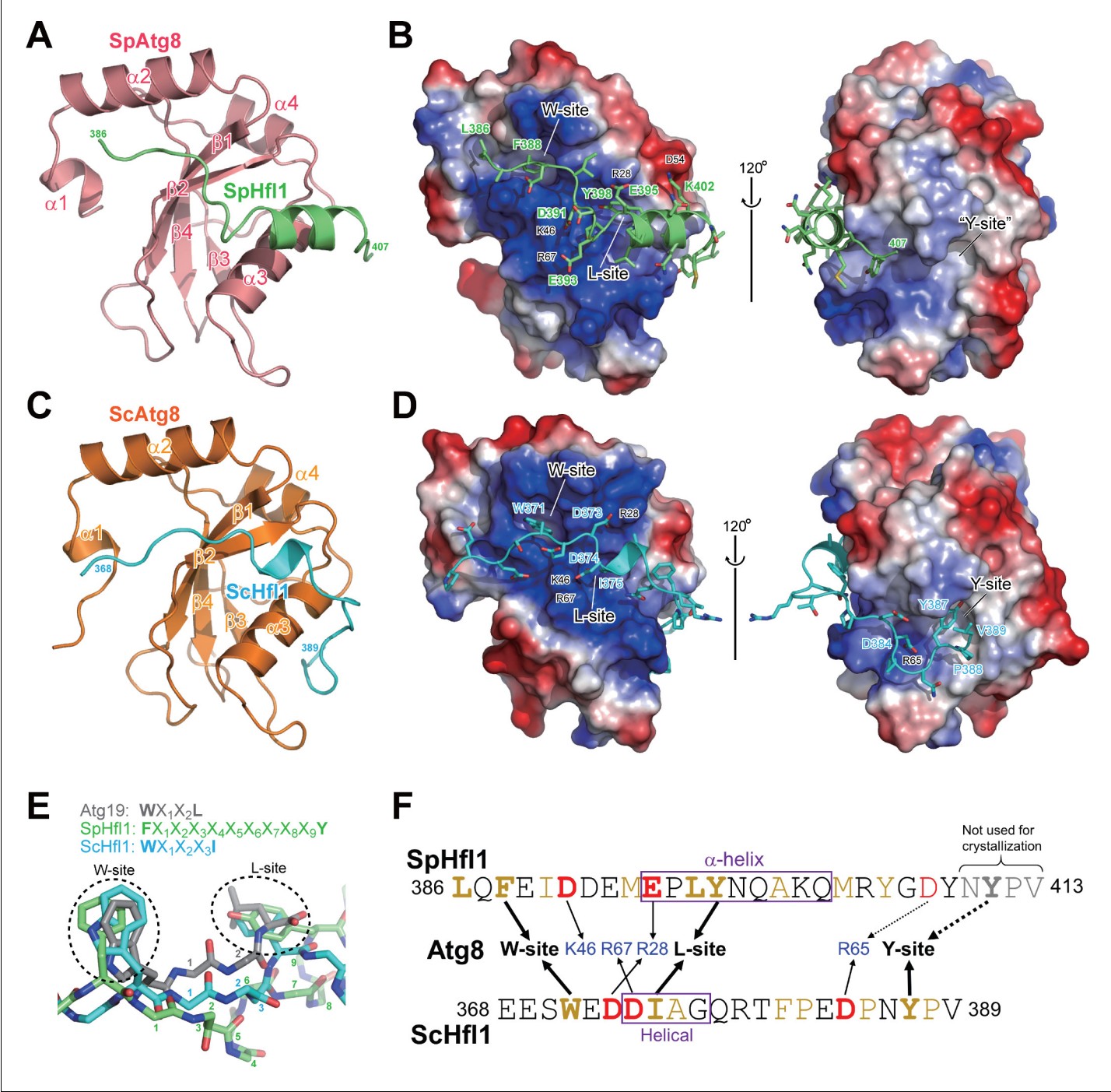

**Figure 3.** Atg8-Hfl1 binding is mediated by noncanonical AIMs and involves a previously unknown binding site on ScAtg8. (**A, C**) Overall structures of the SpAtg8-SpHfl1(386-409) complex (**A**) and the ScAtg8-ScHfl1(368-389) complex (**C**) in ribbon diagrams. (**B, D**) Electrostatic surface potentials calculated for the surfaces of SpAtg8 (**B**) and ScAtg8 (**D**). (**E**) Structural comparison of a canonical AIM and the helical AIMs in SpHfl1 and ScHfl1. Crystal structures of ScAtg8-ScAtg19 (PDB 2ZPN; a representative structure of canonical AIMs bound to Atg8), SpAtg8-SpHfl1 and ScAtg8-ScHfl1 complexes were superimposed with each other by minimizing the rms differences of main-chain atoms of Atg8. The main chains of ScAtg19, SpHfl1, and ScHfl1 are shown with stick models, and the side chains of the two hydrophobic residues that bind to the W-site and the L-site are also shown. The residues intervening the two hydrophobic residues are numbered. (**F**) Summary of the interactions observed between Atg8 and Hfl1. The residues forming hydrophobic interactions are colored yellow, while those forming electrostatic interactions are colored blue (basic) and red (acidic). Hfl1 residues that when mutated affect the affinity with Atg8 by ITC measurements are highlighted with bold letters.

DOI: https://doi.org/10.7554/eLife.41237.007

*Figure 3 continued on next page*

*Figure 3 continued*

The following figure supplements are available for figure 3:

**Figure supplement 1.** The abilities of SpHfl1 to recruit SpAtg8 and complement the vacuole morphology defect of *Sphfl1Δ* require its amino acids 386–409.

DOI: https://doi.org/10.7554/eLife.41237.008

**Figure supplement 2.** Mapping and structural analysis of the Atg8-binding regions in SpHfl1 and ScHfl1.

DOI: https://doi.org/10.7554/eLife.41237.009

**Figure supplement 3.** LigPlot$^+$ diagrams and crystal packing of SpAtg8-SpHfl1(386-409).

DOI: https://doi.org/10.7554/eLife.41237.010

β3 of ScAtg8, burying ~2400 Å$^2$ of surface area (detailed interactions are summarized in *Figure 3—figure supplement 3A*). As is the case with SpHfl1, ScHfl1 neither forms an intermolecular β-sheet with Atg8 β2 (*Figure 3—figure supplement 2G*, **bottom**), nor uses a canonical AIM sequence for binding to the W-site and the L-site. Instead, ScHfl1(368-389) uses Trp371 at the coil and Ile375 at the short helix for binding to the W-site and the L-site, respectively, with three residues between them (*Figure 3E and F*, *Figure 3—figure supplement 3A*). Unexpectedly, Tyr387 of ScHfl1 binds to a hydrophobic pocket formed between α3 and β3 of ScAtg8. This pocket, which has never been reported to be a binding site for AIMs, is named 'Y-site' because it accommodates a conserved Tyr residue in Hfl1. In addition to the hydrophobic interactions, electrostatic interactions are formed between the side-chains of Asp373, Asp374, and Asp384 of ScHfl1 and Arg28, Arg67, and Arg65 of ScAtg8, respectively, and as many as 12 hydrogen bonds are formed between ScHfl1 and ScAtg8 using both the main-chain and side-chain atoms (*Figure 3—figure supplement 3A*). Because the Atg8-binding regions in both SpHfl1 and ScHfl1 do not form an intermolecular β-sheet with Atg8 (a strictly conserved feature of canonical AIMs) and instead use a helical conformation to bind to the L-site, we named this new type of Atg8-binding sequences 'helical AIMs'.

## Isothermal titration calorimetry (ITC) analysis of helical AIM mutants

To determine the binding affinity of helical AIMs to Atg8, we performed ITC analysis. SpHfl1(386-413) and ScHfl1(362-391), which encompass the Atg8-binding regions in the crystal structures, showed Kd values of 161 nM and 1.38 µM to SpAtg8 and ScAtg8, respectively (*Figure 4A* and *Figure 4—figure supplement 1A*). These affinities, especially that of SpHfl1, are strong compared with canonical AIMs that typically show a Kd value of 1–100 µM (*Zaffagnini and Martens, 2016*). The versions used for crystallographic studies, SpHfl1(386-409) and ScHfl1(368-389), showed only a small decrease in affinity (327 nM and 2.20 µM, respectively), whereas further truncations markedly reduced the affinities, confirming that the regions of Hfl1 used for structural studies are necessary and sufficient for strong Atg8 binding (*Figure 4A* and *Figure 4—figure supplement 1A*). Next, we designed helical AIM mutants to validate the interactions observed in the crystal. Alanine-substitution of SpHfl1 Phe388 (F388A) and Tyr398 (Y398A), which bind to the W-site and the L-site in SpAtg8, resulted in ~4 fold and ~1000 fold decrease in the affinity with SpAtg8, respectively (*Figure 4A*). This result suggests that binding to the L-site is much more important than that to the W-site for the SpHfl1-SpAtg8 interaction, which is in contrast to the interactions between canonical AIMs and Atg8 where the W-site binding is usually more important than the L-site binding (*Noda et al., 2008*). In this regard, ScHfl1 is more similar to canonical AIMs than SpHfl1: ScHfl1 uses Trp371 and Ile375 for binding to the W-site and the L-site, respectively, and the W371A mutation showed a much more pronounced reduction in affinity with ScAtg8 (~170 fold) than the I375A mutation (~13 fold) (*Figure 4A*). Canonical AIMs often contain acidic residues between or upstream of the two hydrophobic residues. Alanine substitution of several acidic residues located between the two hydrophobic residues of Hfl1 (D391A and E395A in SpHfl1, and D373A and D374A in ScHfl1) moderately reduced the affinity with Atg8 (*Figure 4—figure supplement 1B*), indicating that electrostatic interactions also contribute to the affinity to some extent. Besides the residues binding to the canonical binding surfaces of Atg8, alanine substitution of Tyr387 and Asp384 in ScHfl1, which bind to the Y-site and Arg65 of Atg8, showed ~5 fold and ~7 fold reduction in affinity, respectively (*Figure 4A* and *Figure 4—figure supplement 1B*). Interestingly, even though SpHfl1(386-409) binds to SpAtg8 almost as strongly as SpHfl1(386-413), in the context of SpHfl1(386-413), the Y411A mutation (corresponding to the Y387A mutation in ScHfl1) resulted in ~24 fold decrease in the affinity

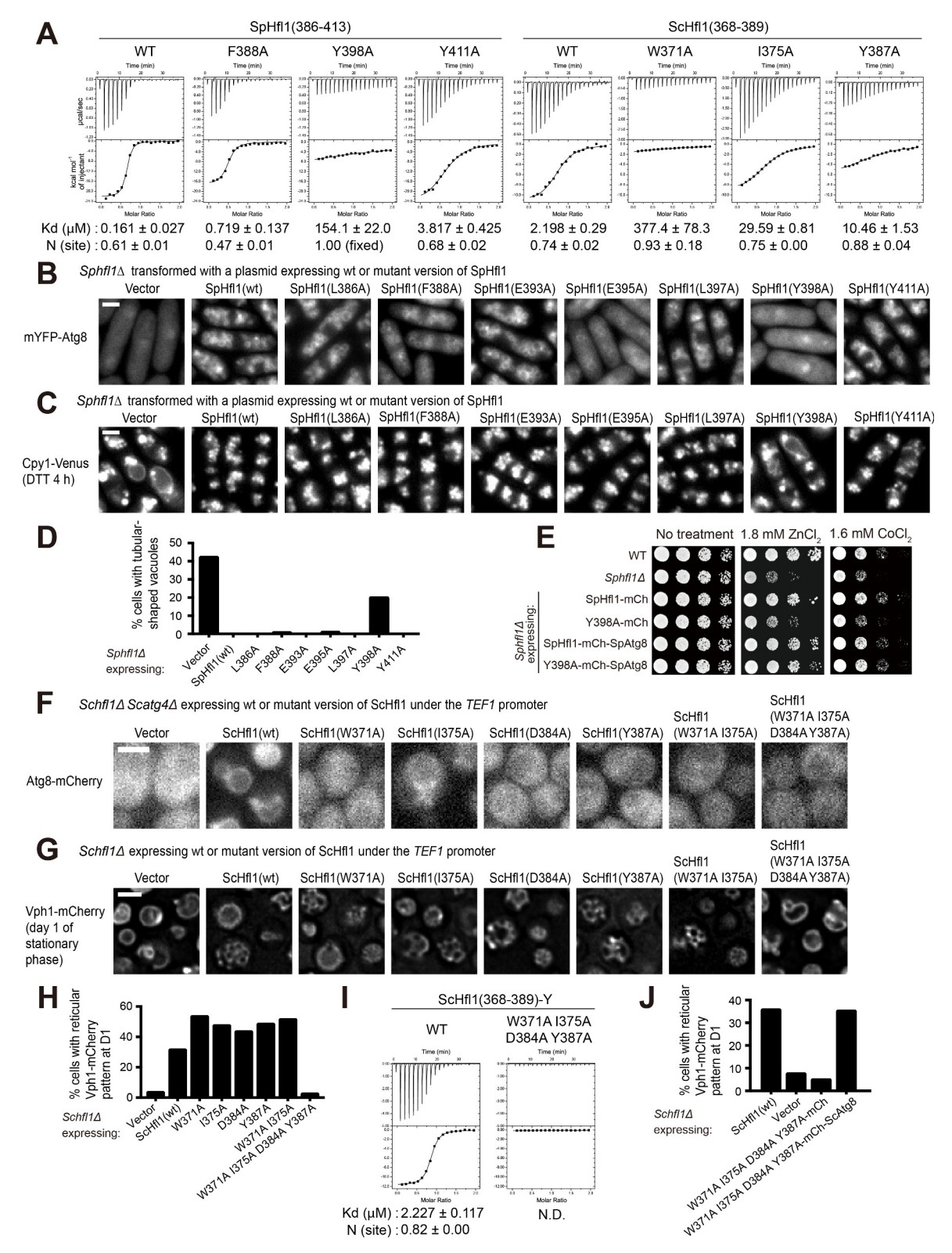

**Figure 4.** The Atg8-Hfl1 interaction is important for the lipidation-independent vacuolar functions of Atg8. (**A**) ITC results obtained by titration of SpHfl1 (386-413) into SpAtg8 or ScHfl1(368-389) into ScAtg8. (**B**) The effects of point mutations in the SpAtg8-binding region of SpHfl1 on the ability of SpHfl1 to recruit SpAtg8 to the vacuole membrane. (**C**) The effects of point mutations in the SpAtg8-binding region of SpHfl1 on the ability of SpHfl1 to complement the vacuole morphology phenotype of *Sphfl1Δ*. (**D**) Quantitation of the vacuole morphology phenotype shown in (**C**). A representative

*Figure 4 continued on next page*

*Figure 4 continued*

result of three independent experiments is shown. (**E**) Y398A mutation strongly diminished the ability of SpHfl1 to complement the metal hyper-sensitivity phenotype of *Sphfl1Δ* and fusion with SpAtg8 restored this ability. (**F**) The effects of point mutations in the ScAtg8-binding region of ScHfl1 on the ability of ScHfl1 to recruit ScAtg8 to the vacuole membrane. ScHfl1 was expressed using the strong *TEF1* promoter. (**G**) The effects of point mutations in the ScAtg8-binding region of ScHfl1 on the ability of ScHfl1 to complement the microdomain formation phenotype of *Schfl1Δ*. ScHfl1 was expressed using the strong *TEF1* promoter. (**H**) Quantitation of the microdomain formation phenotype shown in (**G**). A representative result of three independent experiments is shown. (**I**) ITC results obtained by titration of ScHfl1(368-389) with an additional Tyr residue at the C-terminus into ScAtg8. Addition of Tyr allowed the quantification of the concentration of the ScHfl1(W371A I375A D384A Y387A) by absorbance at 280 nm. (**J**) Fusing ScAtg8 to ScHfl1(W371A I375A D384A Y387A) restored the ability to complement the microdomain formation phenotype of *Schfl1Δ*. A representative result of three independent experiments is shown. Bars, 3 μm.

DOI: https://doi.org/10.7554/eLife.41237.011

The following figure supplements are available for figure 4:

**Figure supplement 1.** ITC experiments for identifying Hfl1 residues important for its binding to Atg8 and structural comparison with two other Atg8 binding sequences that possess a helical conformation.

DOI: https://doi.org/10.7554/eLife.41237.012

**Figure supplement 2.** Fusion with SpAtg8 rescued the ability of SpHfl1-Y398A to complement the vacuole morphology defect of *Sphfl1Δ*.

DOI: https://doi.org/10.7554/eLife.41237.013

**Figure supplement 3.** Fusing SpAtg8 with non-SpHfl1 vacuole membrane proteins did not bypass the requirement of SpHfl1.

DOI: https://doi.org/10.7554/eLife.41237.014

**Figure supplement 4.** Assessing the ability of ScHfl1 mutants expressed from a weak promoter to complement the microdomain formation defect of *Schfl1Δ*.

DOI: https://doi.org/10.7554/eLife.41237.015

with SpAtg8 (*Figure 4A*). Y409A mutation in SpHfl1(386-413) resulted in ~6 fold decrease in the affinity with SpAtg8, whereas the same mutation in SpHfl1(386-409) showed almost no effect on the affinity (*Figure 4—figure supplement 1C*). These observations suggest that additional interactions involving Y411 and Y409 are important for the longer SpHfl1 peptide to bind with SpAtg8 strongly, probably because of the larger entropic cost of the longer peptide (*London et al., 2010*). Thus, it is likely that helical AIMs from both SpHfl1 and ScHfl1 utilize the Y-site of Atg8 for reinforcing the interaction. In sum, ITC data revealed that the L-site-binding Tyr398 is by far the most important residue in SpHfl1 for the SpHfl1-SpAtg8 interaction, while the W-site-binding Trp371 is the most important residue in ScHfl1 for the ScHfl1-ScAtg8 interaction, and that additional hydrophobic and electrostatic contacts play reinforcing roles.

## Atg8-Hfl1 interactions are essential for the lipidation-independent vacuolar functions of Atg8

To determine the functional importance of the Atg8-Hfl1 interactions, we analyzed the in vivo functions of helical AIM mutants. In fission yeast, the point mutations L386A, E393A, L397A, or Y411A affected neither the ability of SpHfl1 to recruit SpAtg8 to the vacuole membrane, nor the ability of SpHfl1 to complement the vacuole morphology defect of *Sphfl1Δ* (*Figure 4B–D*). F388A and E395A moderately weakened the SpAtg8-recruitment ability, but did not substantially affect the ability to complement *Sphfl1Δ*, probably because the residual recruitment of SpAtg8 is largely sufficient for fulfilling the vacuole morphology maintenance function. The point mutation that had the strongest effect in the in vitro ITC analysis, Y398A, completely abolished the SpAtg8-recruitment ability and strongly diminished the ability to complement the vacuole morphology phenotype of *Sphfl1Δ*. Furthermore, we found that SpHfl1(Y398A) lost the ability to complement the metal sensitivity phenotype of *Sphfl1Δ* (*Figure 4E*). To determine whether the phenotype caused by the Y398A mutation is mainly due to a loss of SpHfl1-SpAtg8 association, we performed a fusion rescue analysis. SpHfl1 and SpAtg8 were fused respectively to the N terminus and the C terminus of the same mCherry protein. Because the N and C termini of mCherry are at the same end of the β-barrel structure of mCherry, the spatial proximity of SpHfl1 and SpAtg8 is likely to be preserved. We found that such a fusion restored the ability of SpHfl1(Y398A) to complement *Sphfl1Δ* (*Figure 4E* and *Figure 4—figure supplement 2*), thus demonstrating that the phenotypic consequences of the Y398A mutation mainly result from a disruption of the SpHfl1-SpAtg8 binding. Together, these results indicate that

the physical association between SpHfl1 and SpAtg8 is essential for the lipidation-independent vacuolar functions of SpAtg8.

To assess whether the role of SpHfl1 is solely the recruitment of SpAtg8 to the vacuole membrane, we applied the fusion approach to determine whether we can bypass the requirement of SpHfl1 by fusing SpAtg8 to other vacuole membrane proteins with a C-terminal cytosolic tail. Neither fusion to the vacuolar phytochelatin transporter Hmt1 nor fusion to the vacuolar zinc transporter Zhf1 was able to rescue the vacuole morphology phenotype of *Sphfl1Δ* (*Figure 4—figure supplement 3*), suggesting that SpHfl1 may have role(s) beyond recruiting SpAtg8 to the vacuole membrane.

In budding yeast, consistent with the ITC results, the W371A, I375A, D384A, and Y387A mutations each strongly diminished the ability of *TEF1*-promoter-expressed ScHfl1 to recruit ScAtg8 to the vacuole membrane (*Figure 4F*). Surprisingly, none of these mutations individually had an obvious effect on the ability of ScHfl1 expressed from either the strong *TEF1* promoter or the weak *URA3* promoter to complement the microdomain formation defect of *Schfl1Δ* (*Figure 4G and H*, and *Figure 4—figure supplement 4*), suggesting that the residual Atg8-binding ability is sufficient for the microdomain formation function. To further reduce the ScAtg8-binding ability of ScHfl1, we combined mutations together and found that combining W371A and I375A to simultaneously disrupt the interactions at the W-site and the L-site was not able to abolish the ability of ScHfl1 to complement *Schfl1Δ*, whereas combining W371A, I375A, D384A, and Y387A together led to the complete loss of the complementing ability (*Figure 4G and H*, and *Figure 4—figure supplement 4*). We confirmed by ITC measurements that the double mutant (W371A I375A) retained weak affinity, but that the quadruple mutant (W371A I375A D384A Y387A) showed no binding with ScAtg8 (*Figure 4I* and *Figure 4—figure supplement 1B*). Thus, ScAtg8-ScHfl1 interactions outside of the W-site and the L-site are sufficient for supporting the microdomain formation function. The ability of ScHfl1(W371A I375A D384A Y387A) to complement *Schfl1Δ* was restored by fusing it with ScAtg8 (*Figure 4J*), indicating that the microdomain formation defect of this ScHfl1 mutant is indeed due to a loss of ScAtg8-binding ability. In sum, our results showed that the physical interaction between Atg8 and Hfl1 is vitally important to the lipidation-independent vacuolar functions of Atg8 in these two model yeast species.

## Discussion

In this study, we discovered the molecular underpinnings of the lipidation-independent vacuolar functions of Atg8 in yeasts. Our results provide new insights on two fronts: on the one hand, we identified the integral membrane protein Hfl1 as a key partner of Atg8 in fulfilling its lipidation-independent vacuolar functions, and thus opening up the 'black box' in this special aspect of Atg8 functions; on the other hand, we found that, unlike canonical Atg8 binders, Hfl1 proteins employ a heretofore-unknown Atg8-binding mode—helical AIMs.

Canonical AIMs/LIRs share two common features: one is that they adopt an extended β-conformation and form an intermolecular β-sheet with the β2 strand of Atg8, and the other is that they use the two conserved hydrophobic residues in the W/F/YxxL/I/V motif to bind to the W- and L-sites, respectively, on Atg8. The helical AIMs we identified in SpHfl1 and ScHfl1 do not conform to these rules: they do not form an intermolecular β-sheet with Atg8 and they do not have a W/F/YxxL/I/V motif. Instead, SpHfl1 and ScHfl1 respectively use FxxxxxxxxxY and WxxxI sequences for Atg8 binding. The larger numbers of intervening residues between the two hydrophobic residues are possible because helical AIMs do not have to form an intermolecular β-sheet—if an intermolecular β-sheet is formed, the number of intervening residues must be strictly two.

Other than the helical AIMs described here, there have been two reports describing Atg8 binding sequences that possess a helical conformation and do not form an intermolecular β-sheet with Atg8 (*Figure 4—figure supplement 1D*). One example is a non-natural synthetic peptide named K1 that binds to GABARAP with high affinity (Kd = 354 nM) (*Weiergräber et al., 2008*). The K1 peptide contains a WxxLxW sequence, and interestingly the sequence binds to GABARAP in a reverse direction: the N-terminal WxxL portion with a helical conformation binds to the L-site using both Trp and Leu, while the C-terminal Trp in the sequence binds to the W-site. It remains to be elucidated whether such binding sequence is present in natural proteins. Another example is the coiled-coil region of the retroviral restriction factor Trim5α, which was reported very recently to bind to mammalian

ATG8 proteins using the coiled-coil architecture, and thus is named helical LIR (*Keown et al., 2018*). Helical LIR of Trim5α uses Trp196 to bind to the W-site, but lacks a hydrophobic residue that binds to the L-site, which may be one reason for its weak affinity with mammalian Atg8 homologs (Kd = 80 ~ 100 μM). The functional importance of helical-LIR-mediated Trim5α-Atg8 interaction remains unclear.

The helical AIMs described here are distinct from canonical AIMs with an accessory helix such as the AIM/LIR in FYCO1 (*Cheng et al., 2016*), which uses a canonical W/F/YxxL/I/V motif for interaction and forms an intermolecular β-sheet with Atg8. In FYCO1, a helix is attached to the canonical AIM and increases the affinity by additional interactions. Extended AIMs/LIRs in ankyrin proteins also possess a helix attached to a canonical AIM and show exceptionally strong affinity to mammalian Atg8 family proteins (*Li et al., 2018*). The noncanonical LIR motif of NDP52 (termed CLIR) uses LVV sequence for specific interaction with LC3C, a mammalian Atg8 homolog (*von Muhlinen et al., 2012*). Although the sequence is quite distinct from canonical AIMs, CLIR also forms an intermolecular β-sheet with LC3C in a manner similar to canonical AIMs.

In canonical AIMs, the aromatic residue that binds to the W-site of Atg8 is essential for the interaction—without it almost no interaction is observed. In the case of helical AIMs, the aromatic residue that binds to the W-site can be much less important: the F388A mutation in SpHfl1 only mildly reduced the binding affinity with SpAtg8. Although the W371A mutation in ScHfl1 severely reduced the binding affinity with ScAtg8, it did not disrupt the function of ScHfl1 in maintaining vacuole morphology. Usage of the 'Y-site' as another binding site may contribute to the low dependence on W-site.

Another extraordinary characteristic of helical AIMs is that Tyr can be used to bind to the L-site of Atg8. In the case of canonical AIMs, Leu, Ile, and Val are favored L-site-binding residues because the distance between the L-site and the L-site-binding residue is restrained owing to the intermolecular β-sheet, which makes it difficult for larger residues such as Tyr to bind to the L-site without steric clash. In contrast, in the case of helical AIMs, the distance between the L-site-binding residue and the L-site can be optimized by repositioning the helix, thus allowing a large residue to fit the L-site.

Both Atg8 and Hfl1 are conserved eukaryotic proteins. Yeast Hfl1 proteins closely resemble the mammalian TMEM184 proteins in the seven-transmembrane helix region. TMEM184A (also called Sdmg1) localizes to endosomes in mouse Sertoli cells and is required for the normal localization of the plasma membrane SNARE protein Stx2 (*Best et al., 2008*). TMEM184A also acts as a heparin receptor in vascular cells and regulates angiogenesis (*Farwell et al., 2017*; *Pugh et al., 2016*). TMEM184B localizes to recycling endosomes in mouse neurons and is important for the maintenance of synaptic architecture (*Bhattacharya et al., 2016*). In the model plant *Arabidopsis thaliana*, two TMEM184-related proteins, LAZ1 and LAZ1H1, localize to the vacuole membrane and are redundantly required for normal vacuole morphology and functions (*Liu et al., 2018*). Thus, yeast Hfl1 proteins and their closest homologs in animals and plants share a common attribute of localizing to either endosomal or lysosomal compartments, but whether they share a common function remains to be determined.

More distantly related to the yeast Hfl1 proteins but belonging to the same InterPro IPR005178 protein family are the animal SLC51A/OSTα proteins, which bind to the single-transmembrane SLC51B/OSTβ proteins and transport bile acids and steroids at the plasma membrane (*Ballatori et al., 2013*; *Dawson et al., 2010*). It is tempting to speculate that like SLC51A proteins, Hfl1 and TMEM184 proteins may also act as transporters. However, a recent phylogenetic study has grouped TMEM184 proteins and SLC51A proteins into the transporter–opsin–GPCR (TOG) superfamily, which includes non-transporter proteins such as the G protein-coupled receptors (GPCRs) (*Saier et al., 2016*; *Yee et al., 2013*), hinting that the InterPro IPR005178 family proteins may not necessarily all be transporters. It is possible that Hfl1, as a GPCR-like transmembrane protein on the vacuole membrane, could play a sensing role to monitor changes occurring inside the vacuole lumen, and Atg8 may facilitate signal transduction in the cytoplasm.

The two vacuole-related phenotypes of *Spatg8Δ* and *Sphfl1Δ* mutants—metal sensitivity and abnormal vacuole morphology—may share the same underlying cause, as the same perturbation of vacuole organization or function may reduce the ability of this organelle to sequester metals, and at the same time, alter its morphology. Consistent with this idea, genome-wide deletion library screens in *S. cerevisiae* have shown that a substantial fraction of deletion mutants sensitive to metals are those exhibiting abnormal vacuole morphology (*Pagani et al., 2007*; *Ruotolo et al., 2008*). Even

though vacuole morphology maintenance has not been as extensively investigated in *S. pombe* as in *S. cerevisiae*, the tubular vacuole morphology phenotype of *Spatg8Δ* and *Sphfl1Δ* mutants seems quite distinctive. To our knowledge, the only other genetic perturbation that can result in tubular vacuoles in fission yeast is the overexpression of the dynamin homolog Vps1 (*Röthlisberger et al., 2009*). Furthermore, *vps1Δ* suppressed the tubular vacuole phenotype of *Spatg8Δ* (*Mikawa et al., 2010*). Thus, SpAtg8 and SpHfl1 may directly or indirectly restrain the vacuole tubulation activity of Vps1.

The microdomain formation defect of *Scatg8Δ* and *Schfl1Δ* mutants indicates that ScAtg8 and ScHfl1 contribute to the stationary-phase-induced vacuole membrane partitioning process (*Toulmay and Prinz, 2013*). Previous studies have implicated a diverse group of proteins in this process, including the lipid kinase Fab1, the protein phosphatase Nem1, the MAP kinase Slt2/Mpk1, the ESCRT pathway protein Vps4, the vacuolar protease Pep4, the phosphatidylinositol-3 kinase subunit Atg6, sterol ester synthesis enzymes Are1 and Are2, and sterol transport proteins Lam6/Ltc1, Ncr1, and Npc2 (*Murley et al., 2015*; *Toulmay and Prinz, 2013*; *Tsuji et al., 2017*; *Wang et al., 2014*). Further studies will be needed to clarify the mechanism of microdomain formation and determine how ScAtg8 and ScHfl1 are involved.

Lipidation-independent functions of Atg8 in both autophagic and non-autophagic processes have been reported in animals. In *Drosophila*, a form of noncanonical autophagy required for programmed midgut removal is dependent on Atg8 but not Atg7 and Atg3, the E1 and E2 proteins needed for Atg8 lipidation (*Chang et al., 2013*). How Atg8 is recruited to the autophagosomal membrane during this process is unknown. In mammalian cells, a non-autophagic process, the formation of ERAD tuning vesicle/EDEMosome, depends on non-lipidated LC3, but the exact mechanism is unclear (*Calì et al., 2008*; *Reggiori et al., 2010*). Our findings on the lipidation-independent vacuolar functions of Atg8 in yeasts suggest the possibility that interactions between Atg8 and a transmembrane protein may underlie these lipidation-independent functions of Atg8 in animals.

# Materials and methods

### Key resources table

| Reagent type (species) or resource | Designation | Source or reference | Identifiers | Additional information |
|---|---|---|---|---|
| Gene (*Schizosaccharomyces pombe*) | *hfl1* | NA | PomBase: SPAC30D11.06c | |
| Gene (*Saccharomyces cerevisiae*) | *HFL1* | NA | SGD:YKR051W | |
| Genetic reagent (*Schizosaccharomyces pombe*) | Fission yeast strains used in this study | this paper | | See *Supplementary file 2*—Table S2 |
| Genetic reagent (*Saccharomyces cerevisiae*) | Budding yeast strains used in this study | this paper | | See *Supplementary file 2*—Table S3 |
| Antibody | anti-GFP (mouse monoclonal) | Roche | Cat# 11814460001; RRID:AB_390913 | |
| Antibody | anti-mCherry (mouse monoclonal) | Abmart | | |
| Recombinant DNA reagent | Plasmids used for this study | this paper | | See *Supplementary file 2*—Table S4 |
| Commercial assay or kit | GFP-Trap agarose beads | ChromoTek | Cat# gta-20; RRID: AB_2631357 | |
| Commercial assay or kit | Pierce High Capacity Streptavidin Agarose | Thermo Fisher Scientific | Cat# 20359 | |

## Yeast strain construction

Fission yeast and budding yeast strains used in this study are listed in *Supplementary file 2*—Table S2 and S3, respectively, and plasmids used for yeast strain construction are listed in *Supplementary file 2*—Table S4. The deletion strains used in this study were constructed either by PCR amplifying the deletion cassettes in the Bioneer fission yeast deletion strains and transforming

them into our laboratory strains, or by standard PCR-based gene targeting. Strains expressing proteins with tags (mCherry, GFP, Venus, etc.) under native promoters were generated by PCR-based tagging. The strain expressing SpAtg8 fused at the N-terminus with a CFP tag was as previous described (*Sun et al., 2013*). Plasmids expressing proteins under the control of the *Pnmt1* or *P41nmt1* promoter were constructed using modified pDUAL vectors (*Wei et al., 2014*). To construct *atg8-G116A* fission yeast strain, we used overlap-extension PCR to assemble the following three fragments into one final PCR product: the C-terminal region of the *atg8* ORF (amplified using the primer 5'-TTAATTAACCCGGGGATCCGctaaaaaggaaacactgttGcaaat-3', which introduces the G116A mutation, and the primer 5'-acaacacccattgttttgtca-3'), a *kanMX* marker from pFA6a plasmid (amplified using primers 5'- CGGATCCCCGGGTTAATTAA-3' and 5'- CGATGAATTCGAGCTCGTTT-3'), and the sequence downstream of the *atg8* ORF (amplified using primers 5'-AAACGAGCTCGAATTCATCGatcaacaatttgcctgttttaaga-3' and 5'-aaggatagaatcagctgatgat-3'), and transformed the final PCR product into fission yeast. Plasmids expressing proteins in budding yeast under the control of the *pTEF1* promoter were constructed using pNH605 vectors (plasmids cut with PmeI before transformation) (*Zhang et al., 2017*). To create a fission yeast strain expressing Atg8 tagged at its N-terminus with mYFP or mEGFP, *Patg8* promoter, from −609 to +3 bp with respect to the start codon of the *atg8* gene, was cloned between the *Bgl*II and *Pac*I sites of pFA6a-kanMX6-P41nmt1-mYFP or pFA6a-kanMX6-P41nmt1-mEGFP plasmid to replace the *P41nmt1* promoter (*Bähler et al., 1998*). The kanMX6-Patg8-mYFP and kanMX6-Patg8-mEGFP regions of the resulting plasmids were then PCR amplified with primers containing sequences homologous to regions immediately upstream or downstream of the start codon of *atg8* and used to transform a wild type strain JW81 (*h⁻ ade6-M210 leu1-32 ura4-D18*), as previously described (*Bähler et al., 1998*). Colonies resistant to geneticin (G418) were further verified by PCR amplification across the homologous recombination junctions.

## Affinity purification coupled with mass spectrometry (AP-MS) analysis

We used an *Spatg4* deletion mutant fission yeast strain overexpressing SpAtg8 fused at the C terminus with an YFP-FLAG-His$_6$ (YFH) tag to perform affinity purification of SpAtg8. For specificity control, a parallel affinity purification was performed using a fission yeast strain overexpressing YFH-tagged SPBC16E9.02c. About 1500 OD600 units of cells nitrogen-starved for 2 hr were harvested and washed once with ice-cold water and once with ice-cold lysis buffer (50 mM HEPES-NaOH, pH 7.5, 150 mM NaCl, 1 mM EDTA, 1 mM DTT, 10% glycerol). The cell pellet was mixed with equal volume of lysis buffer containing detergent and protease inhibitors (50 mM HEPES-NaOH, pH 7.5, 150 mM NaCl, 1 mM EDTA, 1 mM DTT, 1 mM PMSF, 0.05% NP-40, 10% glycerol, 1 × Roche protease inhibitor cocktail) and 2 × volume ice-cold 0.5 mm glass beads (BioSpec). Cell lysates were prepared by the bead-beating lysis method using a FastPrep-24 instrument at a setting of 6.5 m/s for four cycles of 20 s bead beating and 5 min on-ice cooling. After centrifugation at 13,200 rpm for 30 min twice, the supernatant was incubated with GFP-Trap agarose beads (Chromotek) for 3 hr. After incubation, the beads were washed twice using lysis buffer and twice using lysis buffer without NP-40. Bead-bound proteins were eluted twice by incubation at 65° with elution buffer (1% SDS, 100 mM Tris, pH 8.0). Eluted proteins were precipitated with 20% TCA. Protein precipitates were washed three times using ice-cold acetone and then dissolved in 8 M urea, 100 mM Tris, pH 8.5, reduced with 5 mM TCEP for 20 min, and alkylated with 10 mM iodoacetamide for 15 min in the dark. Then the samples were diluted four folds using 100 mM Tris, pH 8.5, and digested by trypsin (Promega) in 2 M urea, 1 mM CaCl$_2$, 100 mM Tris, pH 8.5. The LC-MS/MS analysis was performed as described previously (*Liu et al., 2015*).

## Antibodies

The antibodies used for immunoblotting were as follows: anti-GFP mouse monoclonal antibody (Roche), anti-mCherry mouse monoclonal antibody (Abmart, Shanghai, China).

## Immunoprecipitation

Cells were lysed in lysis buffer (50 mM HEPES-NaOH, pH 7.5, 150 mM NaCl, 1 mM EDTA, 1 mM DTT, 1 mM PMSF, 0.05% NP-40, 10% glycerol, 1 × Roche protease inhibitor cocktail) by bead beating using a FastPrep instrument. After centrifugation at 13,200 rpm for 30 min, the supernatant was

incubated with GFP-Trap agarose beads (Chromotek). After incubation, the beads were washed three times with lysis buffer and bead-bound proteins were eluted using SDS-PAGE sample buffer.

### Fluorescence microscopy

Live cell imaging was performed using a DeltaVision PersonalDV system (Applied Precision) equipped with an mCherry/YFP/CFP filter set (Chroma 89006 set) and a 100 × 1.4 NA objective. Images were acquired with a Photometrics CoolSNAP HQ2 camera or a Photometrics Evolve 512 EMCCD camera, and were analyzed with the SoftWoRx software. For quantitation, at least 140 cells were analyzed for each sample.

### Spot assay

For metal sensitivity analysis, including NaCl, KCl, LiCl, $MgSO_4$, $CaCl_2$, $ZnCl_2$, $CoCl_2$, and $MnCl_2$, five-fold serial dilutions of cells were spotted onto YES solid medium with or without indicated concentration of the chemicals. The plates were incubated for 4 to 6 days at 30°C before scanning.

### CFP-Atg8 processing assay

About 10 OD600 units of yeast cells before and after treatment with nitrogen starvation were harvested and lysed using a post-alkaline extraction method (*Sun et al., 2013*). 10 μl of samples were separated on an SDS-PAGE gel and immunoblotted with anti-GFP antibody.

### Protein expression and purification for peptide pull-down

Plasmids for purification of SpAtg8 and ScAtg8 were generated through In-Fusion cloning technology using pETDuet vector cut with HindIII and EcoRI. $His_6$-tagged SpAtg8 and ScAtg8 were expressed from BL21 *E. coli* cells. After adding 0.4 mM of isopropyl β-D-1-thiogalactopyranoside (IPTG) to induce protein expression, 200 ml of culture was incubated at 18°C for 20 hr. Bacteria cells were lysed in lysing buffer (50 mM phosphate buffer, pH 8.0, 0.3 M NaCl, 10 mM imidazole, 10% glycerol, 1 mM PMSF) by sonication. Purification was performed using Ni-NTA-agarose (QIAGEN). The buffer of the eluate was changed to storage buffer (50 mM Tris-HCl, pH 7.5, 0.1 M NaCl, 10% glycerol).

### Peptide pull-down assay

30 μg of recombinant protein purified from *E. coli* and 2 μg of biotin-labeled peptide (1 mg/ml, GenScript) were mixed together in binding buffer (50 mM Tris-HCl, pH 7.5, 0.1 M NaCl, 0.05% NP-40, 1 mM PMSF) and incubated with rotation at 4°C for 3 hr. 15 μl of streptavidin agarose beads (Thermo) was added and incubated for 1 hr at 4°C. Beads were washed with 1 ml of binding buffer for four times. Bead-bound protein was eluted by SDS sample buffer. Samples were analyzed by SDS-PAGE and Coomassie Brilliant Blue (CBB) staining.

### Plasmids for expressing recombinant protein used in structural analysis

All mutations were generated by PCR-based mutagenesis. SpAtg8 gene optimized for bacteria expression was purchased from GenScript. cDNA encoding ScAtg8(1-116) with K26P mutation and the synthesized gene encoding wild-type SpAtg8(1-116) were inserted into pGEX-6P-1 (GE Healthcare) with NdeI and BamHI. As a result, artificial Gly-Pro-His residues were generated ahead of the original first Met of these proteins after GST-tag removal. ScHfl1 variants and ScHfl1(368-389)-ScAtg8 K26P fusion were cloned into the downstream of human rhinovirus (HRV) 3C recognition sequence (HRV 3C seq) of pGEX-6P-1 by using NEBuilder HiFi DNA Assembly Master Mix (New England Biolabs). MBP expression vector was based on a pET15b vector in which HRV 3C seq and MBP gene were inserted with NcoI and NdeI. SpHfl1 variants were cloned into the upstream of HRV 3C seq in the MBP expression vector. As a result, artificial Met and Leu-Glu-Val-Leu-Phe-Gln were added to N- and C-termini of SpHfl1 variants, respectively.

### Protein expression and purification for structural analysis

All proteins for crystallizations and in vitro experiments were expressed in *E. coli* BL21 (DE3). After cultivating bacteria at 37°C until $OD_{600}$ reached 0.8 to 1.2, overnight culturing with 100 μM IPTG was performed at 16°C. After centrifugation, the bacteria were resuspended to PBS with 5 mM

EDTA and lysed by sonication for 10 min. After centrifugation, the supernatants were incubated with affinity resin column: GST accept resin (Nacalai Tesque) for GST-fused proteins and Amylose Resin High Flow resin for MBP-fused proteins (New England Biolabs). After washing the resin with PBS three times, the proteins were eluted with glutathione buffer (10 mM glutathione and 50 mM Tris-HCl pH 8.0) for GST-fused proteins or maltose buffer (10 mM maltose, 20 mM Tris-HCl pH 8.0, 200 mM sodium chloride) for MBP-fused proteins. The eluates were then digested by HRV 3C protease at 4°C for overnight to remove the affinity tag. The proteins were further subjected to size exclusion chromatography (SEC) with 20 mM HEPES pH 6.8 and 150 mM sodium chloride by using Superdex 75 26/60 or Superdex 75 10/300 column (GE Healthcare). Synthesized SpHfl1(386-398) peptide (purchased from Bex Co.) was dissolved in water and purified by SEC with 20 mM HEPES pH 6.8 and 150 mM sodium chloride by using Superdex peptide 10/300 column (GE Healthcare).

## Crystallization

All crystallization trials were performed by the sitting-drop vapor-diffusion method. Protein and reservoir solutions were mixed at 1:1 vol ratio and equilibrated against the reservoir solution at 20°C. To crystallize the complex of SpHfl1(386-409) with SpAtg8, 4.5 mg/ml of SpHfl1(386-409) and 5.9 mg/ml of SpAtg8 were mixed at a molar ratio of 1:1 and incubated for 1 hr at 4°C. The crystals were obtained after 24 hr incubation using 27.5% PEG8000, 0.2 M sodium acetate, 0.1 M Bis-Tris pH 5.5 as a reservoir solution. To crystallize ScHfl1(368-389)-ScAtg8 K26P fusion proteins, 50.3 mg/ml of the fusion protein was used. The crystals were obtained after 24 hr incubation using 6% PEG6000, 0.1 M sodium citrate pH 5.0.

## Diffraction data collection

The crystals of SpHfl1-SpAtg8 complex were soaked in the reservoir supplemented with 5% MPD and transferred to liquid nitrogen. The ScHfl1 crystals were sequentially soaked in the reservoir supplemented with 75 mM sodium chloride and 10, 20, or 27% glycerol and transferred to liquid nitrogen. The flash-cooled crystals were kept in a stream of nitrogen gas at −178°C during data collection. Diffraction data of the SpHfl1-SpAtg8 crystals and the ScHfl1-ScAtg8 crystals were collected by using EIGER X 9M detector at the beamline of BL32XU, SPring-8, Japan and ADSC Quantum 315 r detector at the beamline of BL-5A, KEK, Japan, respectively. The diffraction data were indexed, integrated, and scaled using XDS (*Kabsch, 2010*) for the SpHfl1-SpAtg8 crystals and the HKL2000 program suite (*Otwinowski and Minor, 1997*) for the ScHfl1-ScAtg8 crystals.

## Structure determination

The structures of the SpHfl1-SpAtg8 complex and the ScHfl1-ScAtg8 fusion protein were solved by the molecular replacement method with the program Phenix (*Adams et al., 2010*). For both structures, the crystal structure of ScAtg8 (*Noda et al., 2008*) (PDBID: 2ZPN) was used as a search model. Crystallographic refinement was performed with Phenix. Manual model building was done with the COOT program (*Emsley et al., 2010*). Ramachandran plot analysis with the program Rampage (*Lovell et al., 2003*) showed that 96.9% and 3.1% residues of the SpAtg8-SpHfl1 complex structure and 97.1% and 2.9% residues of the ScAtg8-ScHfl1 complex structure are in the favored and allowed regions, respectively. All structural models in this manuscript were prepared with the program PyMOL except for those with electron-density map (*Figure 3—figure supplement 2E and I*), which were prepared with COOT. Superimposition of structures in *Figure 3E*, and *Figure 3—figure supplement 2F and K* was performed by minimizing the rms difference of main-chain atoms of Atg8 using COOT.

## 2D protein interaction diagrams (LigPlot⁺ diagrams)

Diagrams were generated with PDB ID 6AAF (chain A and B) and 6AAG (chain A and F) using LigPlot⁺ ver 2.1 (*Laskowski and Swindells, 2011*). Hydrogen-bond calculation parameters were set to 2.70 and 3.35 as maximum H-A and D-A distances, respectively. Non-bonded contact parameters were set to 2.90 and 3.90 as minimum and maximum contact distances, respectively. Representative-hydrophobic-only option was used for clarity.

## GST pulldown assay

GST pulldown assay was performed as previously described (*Yamasaki et al., 2016*). Briefly, 50 µg of GST-fused ScHfl1 variants were incubated with 7.5 µl of GST-accept resin in 300 µl of PBS for 1 hr. After short centrifugation, the supernatants were removed, and 50 µg of ScAtg8 were added with 300 µl of PBS. The resin was incubated for 60 min, washed three times with PBS, and eluted with glutathione buffer. Sample buffer was added to the eluate and boiled. Samples were analyzed by SDS-PAGE and stained with CBB. The gel images were captured by Gel-Doc EZ (Bio-rad).

## Isothermal titration calorimetry

ITC experiments were done using Microcal iTC200 calorimeter (Malvern Panalytical), with stirring at 1000 rpm at 25℃. Hfl1 peptides and Atg8 were prepared at the concentrations of 250 µM and 25 µM for SpHfl1(386-413) WT data in *Figure 4A* and all data in *Figure 4—figure supplements 1A*, 300 µM and 30 µM for SpHfl1(386-413) variants and ScHfl1(368-389) WT and Y387A data in *Figure 4A* and SpHfl1(386-413) variants and ScHfl1(368-389) D384A data in *Figure 4—figure supplement 1B*, or 2 mM and 200 µM for ScHfl1(368-389) W371A and I375A data in *Figure 4A*, all data in *Figure 4I*, and ScHfl1(368-389) D373A, D374A and W371A I375A data in *Figure 4—figure supplement 1B*, respectively. 2 µl of Hfl1 peptides in the syringe were injected into a sample cell filled with 200 µl Atg8 for 18 times at intervals of 120 s. The same set of syringe samples were also titrated to a sample cell filled with 200 µl buffer and the obtained reference data were used for subtraction of heat of dilution. MicroCal Origin 7.0 software was used to determine the enthalpy (ΔH), dissociation constant (Kd) and stoichiometry of binding (N). Thermal titration data were fit to a single-site binding model, and thermodynamic parameters ΔH and Kd were obtained by fitting to the model. When the fitting was not convergent due to weak interaction, N was fixed to 1.0 in order to acquire values of other parameters. The error of each parameter shows the fitting error.

## Data availability

The atomic coordinates and reflection data of the crystal structures of fission yeast and budding yeast Atg8-Hfl1 complexes have been deposited in the Protein Data Bank under accession codes 6AAF and 6AAG, respectively.

## Acknowledgments

We thank Chao-Wen Wang for providing *S. cerevisiae* strains expressing Vph1-mCherry, Zhi-Ping Xie for providing *S. cerevisiae* strains expressing GFP-ScAtg8 and helpful suggestions, and Ping Wei for the pNH605 vectors. This work was supported by JSPS KAKENHI (25111004, 18H03989 to NNN; 17K18339 to AY) and CREST, Japan Science and Technology Agency (JPMJCR13M7 to NNN), by National Institute of General Medical Sciences of NIH grant R01GM118746 to JQW, and by funding to LLD from the Ministry of Science and Technology of China.

## Additional information

#### Competing interests

Hitoshi Nakatogawa: Reviewing editor, *eLife*. The other authors declare that no competing interests exist.

#### Funding

| Funder | Author |
| --- | --- |
| Japan Society for the Promotion of Science | Akinori Yamasaki Nobuo N Noda |
| Japan Science and Technology Agency | Nobuo N Noda |
| National Institute of General Medical Sciences | Jian-Qiu Wu |

| | |
|---|---|
| Ministry of Science and Technology of the People's Republic of China | Li-Lin Du |

The funders had no role in study design, data collection and interpretation, or the decision to submit the work for publication.

## Author contributions

Xiao-Man Liu, Akinori Yamasaki, Xiao-Min Du, Investigation, Writing—original draft; Valerie C Coffman, Jian-Qiu Wu, Resources, Writing—review and editing; Yoshinori Ohsumi, Hitoshi Nakatogawa, Resources; Nobuo N Noda, Li-Lin Du, Conceptualization, Writing—original draft, Writing—review and editing

## Author ORCIDs

Hitoshi Nakatogawa ⓘ https://orcid.org/0000-0002-5828-0741
Nobuo N Noda ⓘ https://orcid.org/0000-0002-6940-8069
Li-Lin Du ⓘ http://orcid.org/0000-0002-1028-7397

## Decision letter and Author response

Decision letter https://doi.org/10.7554/eLife.41237.024
Author response https://doi.org/10.7554/eLife.41237.025

# Additional files

## Supplementary files

• Supplementary file 1. Data of the affinity purification coupled with mass spectrometry (AP-MS) analysis.
DOI: https://doi.org/10.7554/eLife.41237.016

• Supplementary file 2. Supplementary tables listing the crystallographic data collection and refinement statistics, and the yeast strains and plasmids used in this study.
DOI: https://doi.org/10.7554/eLife.41237.017

• Transparent reporting form
DOI: https://doi.org/10.7554/eLife.41237.018

## Data availability

The atomic coordinates and reflection data of the crystal structures of fission yeast and budding yeast Atg8-Hfl1 complexes have been deposited in the Protein Data Bank under accession codes 6AAF and 6AAG, respectively.

The following datasets were generated:

| Author(s) | Year | Dataset title | Dataset URL | Database and Identifier |
|---|---|---|---|---|
| Yamasaki A, Noda NN | 2018 | Crystal structure of fission yeast Atg8 complexed with the helical AIM of Hfl1 | http://www.rcsb.org/structure/6AAF | Protein Data Bank, 6AAF |
| Yamasaki A, Noda NN | 2018 | Crystal structure of budding yeast Atg8 complexed with the helical AIM of Hfl1 | http://www.rcsb.org/structure/6AAG | Protein Data Bank, 6AAG |

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
