## [Decision Letter]

Thank you for submitting your article "Lipidation-independent functions of Atg8 rely on its noncanonical interaction with vacuole membrane protein Hfl1" for consideration by *eLife*. Your article has been reviewed by three peer reviewers, including Noboru Mizushima as the Reviewing Editor and Reviewer #1, and the evaluation has been overseen by Andrea Musacchio as the Senior Editor.

The reviewers have discussed the reviews with one another and the Reviewing Editor has drafted this decision to help you prepare a revised submission.

Summary:

Atg8 is one of the autophagy proteins and is covalently conjugated with phosphatidylethanolamine to promote autophagy. However, it has been known that Atg8 has autophagy- and lipidation-independent functions. This study identifies a novel Atg8-binding vacuolar protein, Hfl1, and shows that it is required for the lipidation-independent functions of Atg8 at the vacuole such as maintenance of vacuolar morphology and metal resistance. Furthermore, this study determines the crystal structure of Atg8 in complex with the Atg8-interacting region of Hfl1 and reveals a novel binding motif in Hfl1, which is termed "helical Atg8-family-interacting motif (AIM)" and distinct from canonical AIM. These findings are mostly conserved in the two yeast species *Schizosaccharomyces pombe* and *Saccharomyces cerevisiae* and may be applied to other organisms including mammals. This is a well-organized study addressing an important question in the field.

Essential revisions:

1) The authors suggest that the Y-site in Hfl1 is important for interaction with both ScAtg8 and SpAtg8. In Figure 3B, a clear pocket for accommodating the tyrosine residue is shown in ScAtg8, but it needs to be confirmed also in SpAtg8 using a longer SpHfl1 peptide including Y409 and Y411. In the ITC data, the Y411A mutant peptide shows significantly reduced (~24 fold) affinity (Figure 4A), but SpHfl1(386-409) shows only a modest reduction (~2 fold) (Figure 4—figure supplement 1). This is unreasonable because a point mutant should show a milder effect than a deletion mutant in general. Therefore, it would be important to provide the SpAtg8 structure in complex with a longer peptide. If the experiment takes more than the usual 2-month revision period, at least, the authors should provide the ITC data using the SpHfl1(386-413) Y409A mutant.

2) In general, if a protein-protein interaction is analyzed, a major binding affinity comes from a non-specific interaction between the main chains, and the binding specificity comes from the interaction involving side-chain atoms. However, the current data in Figure 3F does not provide specific interactions at the atomic level and Figure 3—figure supplement 2G does not provide enough information throughout all binding interaction. For clarity of the interaction between Atg8 and Hfl1, it would be nice to include a figure showing detailed interactions such as "Ligplot" for both structures. It will give us more detailed answers regarding the exceptional contribution of the Y398 residue in SpHfl1 to the "L-site" and additional roles of other residues including L386 in SpHfl1 and many acidic residues in both species.

3) The authors show the crystal packing of the ScAtg8-ScHfl1 fusion protein (Figure 3—figure – supplement 2J), but not SpAtg8-SpHfl1(386-409). It would be necessary to show the interaction between the bound SpHfl1 peptide and the symmetry equivalent SpAtg8 protein whether there is an additional force for stabilizing the current structure.

4) In Figure 4F and 4G, what are the expression levels of these ScHfl1 mutants? As shown in Figure 1G, overexpression of Hfl1 could affect the results.

5) The methods and results of the mass spectrometry screen should be included.

---

## [Author Response]

Essential revisions:1) The authors suggest that the Y-site in Hfl1 is important for interaction with both ScAtg8 and SpAtg8. In Figure 3B, a clear pocket for accommodating the tyrosine residue is shown in ScAtg8, but it needs to be confirmed also in SpAtg8 using a longer SpHfl1 peptide including Y409 and Y411. In the ITC data, the Y411A mutant peptide shows significantly reduced (~24 fold) affinity (Figure 4A), but SpHfl1(386-409) shows only a modest reduction (~2 fold) (Figure 4—figure supplement 1). This is unreasonable because a point mutant should show a milder effect than a deletion mutant in general. Therefore, it would be important to provide the SpAtg8 structure in complex with a longer peptide. If the experiment takes more than the usual 2-month revision period, at least, the authors should provide the ITC data using the SpHfl1(386-413) Y409A mutant.

We thank the reviewers for pointing out this important issue. We tried crystallization of both SpAtg8-SpHfl1(386-413) complex and SpHfl1(386-413)-SpAtg8 fusion, but unfortunately failed to obtain crystals. We performed additional ITC experiments using both SpHfl1(386-409) Y409A and SpHfl1(386-413) Y409A mutants and provided the results in Figure 4—figure supplement 1C in the revised manuscript. The new data showed that Y409A markedly affected the affinity of SpHfl1(386-413) to SpAtg8 (from 0.16 μM to 1.01 μM), while scarcely affecting the affinity of SpHfl1(386-409) to SpAtg8 (from 0.33 μM to 0.36 μM). Together with the observation that Y411A mutation significantly reduced the affinity of SpHfl1(386-413) to SpAtg8 while truncation of the 410-413 residues only mildly affected the affinity, we concluded that additional interactions using Y411 and Y409 are important for SpHfl1(386-413), but not for SpHfl1(386-409), to form a strong interaction with SpAtg8. This can be explained by the larger entropic cost of the longer SpHfl1 peptide. We added in the text this explanation and a citation to a paper describing the entropic cost of peptide binding.

2) In general, if a protein-protein interaction is analyzed, a major binding affinity comes from a non-specific interaction between the main chains, and the binding specificity comes from the interaction involving side-chain atoms. However, the current data in Figure 3F does not provide specific interactions at the atomic level and Figure 3—figure supplement 2G does not provide enough information throughout all binding interaction. For clarity of the interaction between Atg8 and Hfl1, it would be nice to include a figure showing detailed interactions such as "Ligplot" for both structures. It will give us more detailed answers regarding the exceptional contribution of the Y398 residue in SpHfl1 to the "L-site" and additional roles of other residues including L386 in SpHfl1 and many acidic residues in both species.

We have performed "Ligplot" analysis for both structures, as suggested by the reviewer. The results are shown in Figure 3—figure supplement 3A of the revised manuscript.

3) The authors show the crystal packing of the ScAtg8-ScHfl1 fusion protein (Figure 3—figure supplement 2J), but not SpAtg8-SpHfl1(386-409). It would be necessary to show the interaction between the bound SpHfl1 peptide and the symmetry equivalent SpAtg8 protein whether there is an additional force for stabilizing the current structure.

The crystal packing of SpAtg8-SpHfl1(386-409) is now shown in Figure 3—figure supplement 3B of the revised manuscript. SpHfl1(386-409) forms little interaction with symmetry-related SpAtg8 molecules, suggesting that the complex structure is not markedly affected by crystal packing.

4) In Figure 4F and 4G, what are the expression levels of these ScHfl1 mutants? As shown in Figure 1G, overexpression of Hfl1 could affect the results.

In both Figure 4F and 4G, we expressed ScHfl1 using the strong *TEF1* promoter, the same promoter used in Figure 1G. We have now clearly stated in Figure 4F and 4G that ScHfl1 was expressed under the *TEF1* promoter. The purpose of the experiments in Figure 4F is to examine whether mutations affect the ability of ScHfl1 to concentrate ScAtg8 on the vacuole membrane. For this assay it is necessary to use a strong promoter to express ScHfl1 because only overexpressed ScHfl1 can exert this effect on ScAtg8. The purpose of the experiments in Figure 4G is to examine whether mutations affect the ability to complement the microdomain formation defect of *Schfl1Δ*. We agree with the reviewers that for this assay the overexpression of ScHfl1 may affect the results. Thus, we have performed new experiments using the weak *URA3* promoter to express ScHfl1. We observed the same results as using the *TEF1* promoter. The new data are shown in Figure 4—figure supplement 4 in the revised manuscript.

5) The methods and results of the mass spectrometry screen should be included.

We have added a description of the affinity purification coupled with mass spectrometry analysis in the Materials and methods section. The results of that analysis are now included as Supplementary file 1 in the revised manuscript.